# Beyond the Game: The Influence of Varying Degrees of Sports Involvement on College Students' Self-Perceptions and Institutional Affiliation

Dalit Lev Arey [1,*] and Orr Levental [2,*]

1    School of Psychology, The Academic College of Tel Aviv Yaffo, Tel Aviv-Yaffo 6818211, Israel
2    Department of Physical Education, Tel Hai Academic College, Qiryat Shemona 1220800, Israel
*    Correspondence: dalitlevarey@gmail.com (D.L.A.); leventalorr@telhai.ac.il (O.L.)

**Abstract:** This research investigates the impact of varying levels of sports participation on college students' self-perception and their sense of connection to their educational institution, with a specific focus on an Israeli context. Recognizing the gap in the existing literature regarding the nuanced effects of different degrees of sports engagement, this study aims to offer insights into how sports involvement shapes students' academic, social, and psychological experiences. Utilizing a qualitative approach, the research involved conducting 26 semi-structured interviews with undergraduate students from a northern Israeli college, encompassing a diverse range of sports participation levels, from occasional involvement to committed varsity athletes. The findings reveal that sports participation significantly enhances students' perceptions of their academic and social abilities, fosters a sense of belonging to the institution, and cultivates institutional pride, regardless of the level of involvement. Particularly noteworthy is the positive correlation between sports engagement and heightened self-esteem and self-efficacy. This study contributes to the understanding of the broader impacts of sports in higher education, highlighting its role in personal development and institutional affiliation. The research underscores the need for further studies in varied cultural contexts to deepen the understanding of these dynamics. Additionally, the study's focus on an Israeli sample provides a unique perspective on the role of sports in a culture where collegiate sports are less academically explored. This research serves as a stepping stone for future quantitative investigations to objectively measure and enhance the understanding of the relationship between sports participation and student development in higher education.

**Keywords:** sports participation; college students; self-perception; institutional pride; higher education; student development

## 1. Introduction

The transformative college years offer myriad opportunities for young adults. Among these, sports participation emerges as a significant and multifaceted experience. While sports culture varies across countries, it remains an essential aspect of university life, providing opportunities for physical, emotional, and social development [1,2]. The existing literature has highlighted the positive impacts of sports participation on academic performance, social integration, and mental well-being [3]. However, there is a gap in understanding how varying degrees of sports involvement influence college students' perceptions and experiences, particularly regarding school identity, self-efficacy, and institutional affiliation [2,4]. This study, focusing on an Israeli sample, aims to fill this gap and potentially provide insights applicable to a broader context.

Unlike previous studies primarily focused on the general effects of sports participation [5], this research delves into the nuanced impacts of varying degrees of sports involvement on college students' attitudes toward their university experience. Our study considers unique cultural elements and the role of sports within universities, exploring

the formation of social connections, overall satisfaction, and perceived academic abilities among students engaged in different levels of sports participation [6,7]. Additionally, we examine the role of sports involvement in fostering institutional pride and a sense of belonging within the university community [8]. We also investigate how the cultural significance of sports, local traditions, and the university's role in promoting sports engagement influence students' identification with their institutions, including the impact of distinct features associated with varsity sports [9,10]. Given its diversified cultural and sports landscape, the Israeli context provides a rich backdrop for this investigation.

## 2. Defining College Sports Participation and Its Forms

College sports participation is a multifaceted concept encompassing a range of physical activities in which students engage within their university environment. These activities typically fall into two categories: occasional sports participation, such as engagement in intramural or club sports, and the highest level of involvement—playing varsity sports as part of the college's teams [9]. Occasional sports participation often involves less commitment and offers students an opportunity to engage in physical pursuits in a less competitive, more inclusive environment [11]. These activities provide diverse benefits, including physical health enhancement, social interaction facilitation, and potential improvement in academic performance through refined time management skills and discipline [12].

In contrast, playing varsity sports represents the highest level of sports involvement. It entails a substantial commitment to a particular sport, often under the aegis of the college's official teams. Varsity athletes adhere to rigorous training schedules, compete regularly against other universities, and may receive sports scholarships, potentially providing a more competitive and high-profile sports experience [13]. This heightened level of engagement often leads students to identify strongly with their athletic roles, significantly influencing their academic, social, and personal experiences [14].

## 3. Social, Psychological, and Well-Being Outcomes of College Sports Involvement

The literature has consistently emphasized the significant impact of sports participation on social, psychological, and well-being outcomes among college students. Sports involvement fosters enhanced social integration by encouraging interaction, teamwork, and a sense of purpose, thus cultivating a sense of belonging and cohesion within the college environment [15,16]. From a psychological standpoint, sports participation is positively correlated with increased self-confidence and enhanced time management skills [17,18]. Soulliard and colleagues [19] provide evidence that student athletes generally display higher levels of self-confidence compared to their non-athlete peers.

Additionally, the structured nature of sports activities necessitates the development of efficient time management skills, a critical component for balancing academic and non-academic responsibilities [20,21]. In terms of overall well-being, sports involvement has been linked to improved mental health outcomes, with students reporting lower stress levels, fewer depressive symptoms, and generally improved mental health [22]. This aspect may be particularly significant within the cultural context of Israel, where mental well-being and holistic development are highly valued [23,24].

## 4. Institutional Loyalty and Belonging: The Role of College Sports Participation

The relationship between college sports participation and students' sense of institutional loyalty and belonging has attracted significant academic attention. As an integral part of college culture, sports profoundly shape students' emotional bonds and overall experiences with their institutions [25]. Substantial research, including that of Hausmann, Schofield, and Woods [26], supports the idea that involvement in sports nurtures a deep sense of belonging among college students. Students who actively participate in sports or are members of athletic teams typically experience heightened feelings of connection to their universities. Furthermore, sports involvement has been associated with increased institutional loyalty. Students who are deeply engaged in college sports often demonstrate

higher levels of loyalty, a sentiment likely rooted in their strong identification with the university—a bond that is strengthened through their participation in sports [27].

**5. College Sports Participation and Self-Esteem and Self-Efficacy**

Self-esteem and self-efficacy, crucial facets of an individual's self-concept, may be significantly influenced by sports participation [28]. Self-esteem relates to an individual's overall subjective emotional evaluation of their worth, while self-efficacy pertains to the belief in their capability to execute tasks and achieve goals [29]. Both constructs are fundamental for personal development and mental well-being in students, playing a vital role in shaping their academic and non-academic performance in a university setting [30,31].

Research has consistently demonstrated that sports participation can significantly influence both self-esteem and self-efficacy. Engagement in sports activities is positively related to increased levels of self-esteem among college students [32]. Similarly, students involved in sports often exhibit higher levels of self-efficacy than those who do not participate [33]. However, the prevailing literature does not adequately differentiate between the effects on self-esteem and self-efficacy among students who participate in sports regularly versus those who engage occasionally. This distinction is important, considering that the intensity, commitment, and competitive nature inherent in these two modes of sports involvement can shape self-perceptions differently. Addressing this gap, our study will explore how varying degrees of sports involvement impact self-esteem and self-efficacy among college students, taking into account the nuances of these different levels of engagement [34].

**6. Research Methodology**

The current research aimed to examine students' attitudes toward the contribution of sports participation in higher education. In pursuit of this goal, the research investigated the perceptions, challenges, and impact of physical activity on students' lives. To achieve this, a qualitative approach was adopted, found to be most suitable for the in-depth exploration of subjective feelings within the phenomenon under investigation [35].

Twenty-six semi-structured in-depth interviews were conducted with undergraduate students from various departments at a college in northern Israel. Participants were recruited through three methods: The first method involved direct outreach to all student athletes who regularly trained with the college's teams. In parallel, an approach was made to students who had participated in specific competitions throughout the year but were not part of the first group (Patch Recruitment). These were students who participated only in competitions but did not regularly engage in team training. The third method was a general call to all students at the academic institution through the college's Facebook page to volunteer for the research. In total, twenty-six students participated in the study: six who regularly participated in college teams, ten who had participated in one or two competitions during the academic year but were not part of the first group, and ten students who did not participate in college sports at all during their studies.

All participants in the study were single, aged between 18 and 27, and resided in various northern Israeli towns, including Kiryat Shmona, Metula, Katsrin, Kfar Yuval, and Ma'ayan Baruch, among others. Most interviews were conducted face-to-face, while some were conducted via the Zoom platform. Face-to-face interviews were held in quiet, isolated locations, allowing participants to freely share their thoughts. Each interview lasted between 30 and 45 min. Subsequently, the interviews were transcribed and underwent thematic analysis conducted independently by two researchers.

The analysis process included the initial coding of meaningful data units, followed by the development of sub-themes and overarching themes. Among the secondary themes that emerged were self-confidence, social relationships, social standing, personal beliefs, competitiveness, commitment, and academic support. These were consolidated into two main themes: perceptions of academic and social competencies, and levels of satisfaction and institutional pride. The findings were analyzed along two primary dimensions: differences among the participant groups and the personal meanings attributed to sports

participation's impact on the broader student experience. Regarding the first dimension, significant differences between the groups were evident, indicating that the extent of sports participation influenced the attitudes and perceptions of the students. In general, occasional sports participation in college improved the students' perception of their abilities, their satisfaction with their studies, and their perceived contribution of sports to these aspects. Furthermore, it should be noted that the research did not examine the objective relationships between the extent of participation and academic achievements or quantitative measures of satisfaction. However, these perceptions were expressed in the interviews as detailed below. Additionally, it is possible that the perceptions described by the participants were what led them to engage in sports as part of their studies. In other words, the aspiration to improve their social situation also included participation in competitive social sports activities.

## 7. Findings

### 7.1. Perception of Abilities—Academic and Social

The research highlights several key findings regarding the impact of sports participation on students' perception of their abilities, both academically and socially. Participants who regularly trained and competed emphasized various aspects, such as motivation and skill development. For instance, Avishai shared his perspective, stating, "I've been training since a young age, I love sports, and it has a central place in my life. Even on a moral level, in terms of competition, and in the importance of the effort to succeed, I've learned to push myself harder, to strive for victory. I don't see any other environment like this". Another notable point in this context is the recognition of the link between investment and success. Tom, for example, articulated, "In sports, you can't cut corners. You can pretend, but not in terms of your results. Those who invest more succeed. It's a thousand times more important than talent. Once you understand that, you bring it to your studies too. Investment in wisdom or knowledge is better".

Interestingly, even among students who participated in sports only once, there was a more positive attitude toward academic success. Mohammad's perspective exemplifies this: "Everyone who makes it to the competition wants to win. The attitude is, you're already here, so give it your maximum. When you return home, you continue. You're learning anyway; at least you'll get good grades". However, it is essential to note that the positive attitude presented by these students may not solely result from sports participation itself but rather from a general perception of it as a positive factor. The differences observed among the groups mainly stem from the fact that students who did not participate in sports at all perceive it as a negative and hindering factor. Uriyah explained this by saying, "They (the athletes in my class) are not here half the semester. One time they play there, another time it's a national championship. So, there's no question that they lack the material and knowledge they need to catch up on. They made a decision that they're involved in sports instead of studying. In the end, they're here for the degree; otherwise, they could have played in professional clubs, it seems like a waste". Interviews with non-sporting participants reveal their difficulty in understanding the potential contribution of sports, viewing it as an extracurricular activity that does not align with higher education goals. Additionally, sports participation requires resources, primarily time, which may not appear, at first glance, to be dedicated to academic efforts.

Another noteworthy aspect that emerged during the interviews was the more positive perception of students engaged in sports regarding their student societal experience. In these cases, the extent of involvement in sports correlated with a more positive attitude towards their student experience and social integration. Tal, for instance, mentioned, "All my closest friends in my studies are from the team (indoor soccer). Add to that the fans who come to the games, all the students who read my name on Facebook and Instagram posts. I'm not just a student". Maya added, "I studied for two years at the college and only knew the girls who studied with me. Then I went to the student tennis competition in Eilat, suddenly I'm part of a group of 40 students. Suddenly, people from other departments in the college are coming to cheer me on, from another campus altogether".

Participants who engaged in sports occasionally or once also highlighted that competitions allowed them to connect with other students at the institution and expand their social circles. Furthermore, they noted that representing the college made them more recognizable to both students and faculty members, fostering a sense of belonging within the institutional framework. However, it is essential to emphasize that students who did not participate in sports struggled to comprehend the potential benefits of sports and, in some cases, perceived it negatively.

Moreover, the research findings suggest that students involved in sports, regardless of the extent of their participation, place greater importance on their social lives as students, in contrast to those who do not participate in sports and primarily prioritize the academic aspect of their studies. For instance, Gil pointed out, "It's not an American movie here. I'm not on vacation for four years of my life. I came to study. In the United States, students are 18-year-old kids, so they come to celebrate and party, and so on. In Israel, it seems to me, people come to study. To get a degree".

From the interviews, it became evident that involvement in sports offers students a fresh perspective on higher education, fostering a more holistic view of their entire college experience. Eilon, a member of the college's running team, noted, "They tell us that the school's role is a high school diploma, the bachelor's degree's role is the diploma. But you can look at it differently. If sports are part of the process, and it's not directly related to my diploma, then maybe we should see this as a broad perspective of experiences". Notably, other students also discovered various aspects of their studies through sports that were connected to their overall student life.

However, students who did not engage in sports, while acknowledging the social aspects of their studies, did not perceive sports activity as an integral or inseparable part of it. When Michal was asked about it directly, she responded, "I don't see how it's really part of the studies. It's like a league for workplaces. It's not part of the work. It's a leisure activity that doesn't really affect the studies". This disparity in their responses and their perception of sports as a separate or non-integral part of the learning process influenced their connection to the institution and their appreciation of the activity within it, as evident in the following section.

### 7.2. Satisfaction and Institutional Pride

Another significant finding emerging from the research pertains to two subthemes: students' satisfaction with the institution and their learning experience. Additionally, the theme of institutional pride emerged, primarily due to their participation in the college's representative sports. As previously discussed, involvement in higher education sports plays a crucial role in shaping students' attitudes towards themselves and the society to which they belong. There was a clear enhancement in both personal and social aspects among students engaged in sports in terms of their overall satisfaction with their studies. For instance, Tom expressed, "It's an experience. It's a wonderful time. I had a dilemma between two educational institutions, and I think I made the right choice. There's a different atmosphere here. I'm also earning a degree, and I'm doing other things". Dror also added, "There are people here who are completely broken. They spend the whole day stressed and looking for what's not good. I don't understand them; maybe it's because their studies are just studies. You can't live just half of the experience. Where are the student days, the sports, the parties?"

It is worth noting that participants in this group expressed overall satisfaction, both with the institution and their student experience, regardless of the extent of their sports involvement. In contrast, those who only participated once during the year did not exhibit significantly different attitudes from those who did not participate at all. It should be emphasized that, in general, the research participants did not express dissatisfaction. Despite differences in their general outlook based on the extent of their sports involvement, they did not express overall dissatisfaction with their degree program. This may be attributed to the relatively low number of hours students invested in occasional events

compared to their investment in courses and exams. Consequently, the group that regularly participated in sports seemed to significantly impact their overall satisfaction, both with the degree and their sporadic experiences within it.

Another subtheme related to the students' connection to the institution is their sense of belonging and a specific focus on their institutional pride as students or graduates of the college. This subtheme encompassed various expressions related to the concept of institutional pride, including recommendations for the institution and wearing clothing adorned with the institution's emblem. It became evident that participation in sports, to some extent, significantly contributes to the development of students' institutional pride. Roni, who participated twice in a soccer competition during his studies (without regular training with other students), shared his perspective: "I didn't think I'd feel anything, but suddenly it's you and your friends with the same clothes, the same color, the same emblem, and there are other teams representing different places. So, the college sending me created some commitment. But even more than that, victory generates pride, and losses make me think that maybe I disappointed someone". In addition to these statements, other interviewees connected their feelings to the nature of sports. The research participants noted that sports create a distinction between groups, and sports affiliation with the college team necessarily leads to affiliation with the college itself.

## 8. Discussion

This research intricately examines the influence of sports participation on holistic aspects of a student's higher education experience and the varied experiences of different participant groups. A significant theme emerging from our findings is the complex relationship between sports engagement, which requires a substantial investment of personal resources, particularly time, and its potential impact on academic success. This dynamic, highlighted by our research, aligns with previous studies that have demonstrated the positive effects of sports on academic achievements through the development of skills such as goal setting and discipline [12]. However, the increased allocation of resources to sports involvement, such as in training and competitions, can lead to a decrease in the time devoted to academic studies [36]. This juxtaposition accentuates the dual nature of sports participation as a facilitator of success and a potential obstacle to academic progress.

Our research indicates that perceptions of sports' effects largely depend on the extent of participation. Students who actively participate often develop an identity associated with the potential benefits, both personally and socially. The findings suggest that extensive sports involvement significantly influences self-perception and the student's role within the institution [13]. Notably, most research on sports in higher education, particularly in the United States, contrasts with other Westernized countries like Israel, where collegiate sports primarily function as recreational extracurricular activities [37]. Our findings propose that students, regardless of their level of participation, perceive sports involvement differently, not viewing it as a potential conflict with academic pursuits. This differentiation reduces the commitment to teams and competitions and mitigates any negative impact on academic aspects. Thus, the absence of significant adverse effects on studies highlights the role of sports as a tool for personal and social development beyond professional sports.

Our study corroborates previous research emphasizing the social, psychological, and well-being benefits of sports among students [15,38]. It indicates that sports encourage social interactions, foster a sense of belonging, and instill important behavioral habits [39,40]. Therefore, participants engaging in sports at any level experience these benefits, contributing to their overall college experience. The self-confidence characteristic of these students [41,42] empowers them to invest more effort in their tasks and strive for success. Importantly, our research focused on students' subjective perceptions rather than objectively measuring the relationship between sports and academic success, reflecting the significance of these perceptions in shaping both their motivation for success and their holistic student experience.

Exposure to sports during their studies leads students to a more comprehensive understanding of the purpose of education [43]. They perceive their studies not just as a means to acquire professional knowledge but as a period for personal development, engagement in recreational activities, and socialization. The extensive involvement in sports, including representing the college at various levels, broadens their perception of education. These students view their studies not merely as a functional path to professional knowledge but as a phase of personal growth, recreational engagement, and social interaction [44]. This perspective underscores the importance of extracurricular social activities in higher education.

*Social Identity and Group Dynamics*

Incorporating Social Identity Theory [45] provides a critical perspective on the impact of sports participation on college students' sense of belonging and institutional affiliation. The theory suggests that part of an individual's self-concept arises from their membership of social groups. In the college context, sports teams and activities represent these groups, offering a shared identity that significantly influences students' self-perceptions and their place within the educational institution. Our study demonstrates that participation in sports, whether casual or competitive, enables students to align with distinct groups within the college community, fostering a deeper sense of belonging and identity. Being part of a sports team or regularly engaging in sports activities often becomes integral to their college identity, consistent with Social Identity Theory, which asserts that group membership enhances self-esteem and belonging [46,47].

The group dynamics in sports, characterized by teamwork and unity, play a crucial role in strengthening institutional affiliation [48]. Students engaging in sports not only interact with their immediate teams but also identify themselves as part of the larger college community [49]. This identification is especially significant in our study, which includes a diverse range of sports participation levels, from occasional club sports to dedicated varsity teams. Each level of involvement contributes to the collective identity and social fabric of the institution.

In Israel, where collegiate sports are less prominent than in countries like the United States and Australia, sports participation assumes a unique role as a unifying social group. The diverse range of sports participation mirrors a multifaceted sports culture within the college, catering to various interests and commitment levels. This diversity is vital in providing multiple avenues for students to engage with and contribute to the college community, reinforcing their sense of belonging and institutional pride [50]. By considering the implications of Social Identity Theory and the dynamics of group membership in sports, we gain a comprehensive understanding of how sports participation influences students' self-perception and institutional affiliation. This theoretical framework enriches our interpretation of the findings and highlights the broader significance of sports as a vehicle for social and psychological development in higher education settings.

## 9. Conclusions

This research delved into exploring the diverse impacts of sports involvement on the self-perception and institutional connection of college students within an Israeli context. Our investigation revealed that regardless of the intensity of sports participation, there is a consistent positive influence on students' perceptions of their academic and social capabilities. Moreover, this involvement cultivates a profound sense of belonging and institutional pride. Notably, we observed that the depth of these benefits in self-esteem and self-efficacy is contingent upon the degree of commitment to sports activities. This highlights a nuanced interplay between sports engagement and personal development, enriching the discourse on the pivotal role of sports in enhancing the higher education experience. Our findings underscore the importance of sports as a catalyst for student growth, weaving a strong fabric of connection with their academic environment.

While our study provides meaningful insights, it is bounded by several limitations. The qualitative nature of our approach, though rich in narrative detail, restricts the breadth of generalizability beyond the 26 participants from a single Israeli college setting. While valuable for understanding personal perspectives, the reliance on self-reported data may introduce subjective biases and limit our capacity to establish definitive causal relationships. Furthermore, the unique cultural nuances of the Israeli academic context might not translate seamlessly to different educational environments, warranting cautious application of our findings in broader contexts.

Our study lays a foundational framework for a range of future research endeavors. We advocate for expansive quantitative research with diverse and larger sample sizes to corroborate and extend our findings. Longitudinal studies would be invaluable in tracing the evolving influence of sports participation throughout a student's academic tenure. Comparative research across various institutional sports cultures could illuminate the differential impacts of sports on student experiences. Additionally, integrating insights from psychology, sports science, and educational theory through interdisciplinary studies could yield a more comprehensive understanding of sports' role in student development. Extending this line of inquiry to various cultural and geographic landscapes will contribute significantly to a global discourse on the impact of sports in higher education, informing educational strategies and policies aimed at fostering well-rounded and socially and academically successful students.

**Author Contributions:** Conceptualization, D.L.A. and O.L.; methodology, O.L.; formal analysis, O.L. and D.L.A.; investigation, O.L. and D.L.A.; writing—original draft preparation, O.L. and D.L.A.; writing—review and editing, O.L. and D.L.A. All authors have read and agreed to the published version of the manuscript.

**Funding:** This research received no external funding.

**Institutional Review Board Statement:** The study was approved by the Ethics Committee of Ohalo College, Israel (1/2020, code01243 ).

**Informed Consent Statement:** Informed consent was obtained from all subjects involved in the study.

**Data Availability Statement:** The data presented in this study are available on request from the corresponding author. The data are not publicly available due to privacy issues.

**Conflicts of Interest:** The authors declare no conflict of interest.

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
