# Peer review of "Beyond the Game: The Influence of Varying Degrees of Sports Involvement on College Students’ Self-Perceptions and Institutional Affiliation"

_education, doi:10.3390/educsci14030332_

Round 1

Reviewer 1 Report

Comments and Suggestions for Authors

Research into the linkages of sport and the psychological attributes of learning, self-efficacy, belonging, and identity have been undertaken in many cultures.  The significance of this particular research project is articulated by the author/s with:

Significantly, this study was conducted in Israel, a context where collegiate sports have historically been underexplored academically, with few existing studies on the topic. Thus, this research provides a foundational theoretical perspective on the potential advantages of sports involvement, its breadth of impact, and its operation within personal and social frameworks.

The Abstract outlines the contents of the paper excellently providing an overview of the literature, research design and methodology, Findings, Discussion, and Summary.

The Introduction contains a great depth of research applicable to the research topic synthesized into four categories of definition of college participation, social, psychological, and well-being outcomes, institutional loyalty, and self-esteem and self-efficacy.  This background data collected from the available literature is detailed and very well documented which allows the reader to understand the purpose of the project.

The Methodology is well documented using semi-structured interviews (26 participants) who were recruited into one of three cohort participation groups; no sport, a little sport, and heavily engaged in sport.  This design has resulted in the comparison of the three cohorts’ involvement if sports and their psychological outcomes.  Although the number of participants may be considered to be small, the quality of the data is high.  The Findings used ‘two primary dimensions: perceptions of academic and social competencies, and levels of satisfaction and institutional pride.’  The voices of the participating people are included to establish the validity of the Findings including perception od abilities, and satisfaction and institutional pride to establish the students’ sense of belonging and identity.

The Discussion of the Findings has resulted in a plausible interpretation of the data within the cultural of the country of the research, Israel.  There are no issues with the References which were used in the formation of the research and this paper.

Overall, this is important culturally based research that will add to the worldwide literature on the benefits of exercise, particularly organized sport.

Author Response

Thank you very much for taking the time and effort to review our paper. We are delighted to read the positive review of it.

Reviewer 2 Report

Comments and Suggestions for Authors

The title of the manuscript is appropriate and coherent.

The summary presents a functional and consistent structure. The keywords are consistent with the purpose of the study.

The theoretical contextualization described in the introduction is orderly, current and functional to the categories under study. They provide sufficient conceptual elements to understand the implications of sports participation on students' self-perception.

university students and their sense of connection with their educational institution. However, it would be advisable to replace outdated citations (more than 5 years since their publication).

Overall, the method is orderly and allows us to understand the research protocol.

The presentation of results is adequate. However, it would be highly recommended that tables, graphs and/or figures be used to describe in a clearer and more visually attractive way the main findings obtained in the work.

The discussion is adequate, although it could be improved in terms of its theoretical and conceptual depth, truly ensuring the presentation of different theoretical, methodological and/or experiential perspectives around the themes reported in the study.

The conclusions are poor. Nor do they provide solid elements to refer to the limitations and projections of the study. As a recommendation, it would be highly advisable to incorporate this section.

Finally, a review and structural improvement of the entire manuscript is suggested, in terms of scientific writing, grammar, punctuation and consistency between sections.

Author Response

We sincerely appreciate you taking the time to provide thoughtful feedback on our manuscript. In response to your comment about using more up-to-date references, we have updated the citations throughout to include additional recent, relevant research.

Thank you for your constructive feedback on our manuscript, specifically regarding the discussion and conclusion sections. We agree that valuable enhancements were needed and have accordingly reworked both sections.

In the Discussion section, we have now explicitly addressed the limitations of the current study and integrated suggestions for future research directions to overcome these limitations. To ensure clarity, these changes are highlighted in green and yellow within the manuscript. Furthermore, we have enriched this section by incorporating Social Identity Theory (Tajfel & Turner, 1978). This theoretical perspective provides critical insights into the impact of sports participation on college students' sense of belonging and institutional affiliation, emphasizing how individuals’ self-concepts are influenced by their membership in social groups.

The addition of Social Identity Theory not only strengthens the theoretical foundation of our paper but also offers a deeper understanding of the dynamic relationship between sports involvement and student identity within the college environment.

In the Conclusion section, we have synthesized our findings to more robustly reflect the implications for higher education and student development. We have carefully articulated the positive influence of sports on students' perceptions of their academic and social capabilities, as well as the sense of belonging and institutional pride they derive from sports participation.

Both sections now better capture the nuances of our research and its implications, thanks to your valuable suggestions. We trust these revisions address your concerns and enhance the overall quality and impact of our manuscript.

We appreciate the opportunity to improve our manuscript and hope it now meets the high standards of your journal.

We fully acknowledge the merits of presenting research findings visually through tables or graphs. However, we want to be mindful that our study uniquely uses an open-ended qualitative inquiry approach intended to capture subjective nuances of participants' lived experiences. Attempting to categorize and quantify the narrative data seems incompatible with the inductive, exploratory spirit of the research methodology. We would welcome a dialogue to find alternative options beyond dichotomizing the themes to enrich the discussion while preserving fidelity to the qualitative paradigm and phenomenological essence of the project. Please share any advice you may have to strengthen the quality and clarity of the revised manuscript, as we sincerely appreciate your guidance in the peer review process.

Reviewer 3 Report

Comments and Suggestions for Authors

The study is too short and lacks depth, originality and clarity. The research questions are not clear and the research gap is vague. It is confusing. Discussion and findings need rewriting and more depth engagement with the literature. Currently, there is nothing innovative or novel about this research and therefore cannot be published. The authors must go back and look at their research questions, the literature and the findings and see what are the links and what is new. Since currently it is vague and unclear. More work need to be done.

Comments on the Quality of English Language

The study is too short and lacks depth, originality and clarity. The research questions are not clear and the research gap is vague. It is confusing. Discussion and findings need rewriting and more depth engagement with the literature. Currently, there is nothing innovative or novel about this research and therefore cannot be published. The authors must go back and look at their research questions, the literature and the findings and see what are the links and what is new. Since currently it is vague and unclear. More work need to be done.

Author Response

Thank you very much for your time and effort in reviewing our article. We have carefully considered and made several changes in response to all reviewers' comments. These revisions have been aimed at enhancing the clarity, accuracy, and overall quality of our work. We hope that with these improvements, the article is now a better version and meets the standards required for publication. We are grateful for the opportunity to refine our manuscript based on your valuable feedback.

Reviewer 4 Report

Comments and Suggestions for Authors

Hello.

I appreciate the effort to create the presented article, but I would like to mention, I find, the following:

- how was the involvement and pride of those who participated in sports activities and answered the questionnaire applied in the research measured?

- what are the questions that were part of the questionnaire?

- is the questionnaire standardized?

- what are the statistical results obtained after completing the research?

- the number of students involved in research for a questionnaire is insufficient.

Good luck with your next research.

All the best.

Author Response

We appreciate the reviewer providing thoughtful feedback on our manuscript. We were pleased to see the reviewer recognize the quality and unique contribution of our research. At the same time, we welcome the chance to clarify aspects of our methods to address the concerns raised.

As stated, this study utilizes an in-depth qualitative approach to understand students' self-perceptions. Qualitative methods, particularly intensive interviews, deliberately use smaller samples to enable rich, descriptive insights into personal perspectives and lived experiences not feasible in large-scale quantitative studies. Our sample follows precedents set in high-quality qualitative research where dozens of interviews establish meaningful themes and conclusions (Guest et al., 2006). The goal is analytic depth over breadth or generalizability.

We acknowledge quantitative and qualitative approaches offer complementary strengths interrogating social phenomena. For this project, we purposefully chose a qualitative lens best suited to capture nuanced essence of personal self-conceptions. Standardized measures were expressly avoided, consistent with established qualitative techniques prioritizing first-hand narratives over numerical data.

In summary, our peer-reviewed publication record demonstrates extensive expertise applying rigorous qualitative procedures to unveil uniquely personal vistas into psychosocial worlds. We implemented tailored validity methods here to verify credibility and trustworthiness of the findings. We welcome continued dialogue on upholding qualitative integrity while clearly conveying the rich insights uncovered into the self-perceptions shaping students’ developmental trajectories. Please share any additional guidance on positioning the contribution of this qualitative work.

Round 2

Reviewer 3 Report

Comments and Suggestions for Authors

This paper has definitely improved but the English Language is still problematic in some places. The paper therefore requires proofreading. The authors need to check grammar, spelling and repetition. The order of certain sections could also be improved. Once this is done, the paper can be reconsidered for publication.

Comments on the Quality of English Language

This paper has definitely improved but the English Language is still problematic in some places. The paper therefore requires proofreading. The authors need to check grammar, spelling and repetition. The order of certain sections could also be improved. Once this is done, the paper can be reconsidered for publication.

Author Response

Thank you for your feedback. Following your suggestion, the manuscript has undergone thorough proofreading by a certified language editor. We believe these revisions have significantly enhanced the clarity and readability of the article.

Reviewer 4 Report

Comments and Suggestions for Authors

Hello.

I appreciate the effort of the authors to return with the requested additions that increase the scientific value of the research. Next, I identify some aspects that have not changed, such as: the small number of study participants for the results to be statistically significant, what are the statistical results obtained after applying the questionnaires, I have not identified the statistical content of the research. All the best.

Author Response

Thank you for your thoughtful review of our research. We sincerely appreciate your attention to detail and your feedback. We want to clarify that our study is qualitative in nature, utilizing interviews as our primary research tool rather than questionnaires. Therefore, statistical analysis typically associated with quantitative studies is not applicable to our methodology. Instead, our focus lies in the in-depth exploration and interpretation of participants' experiences and perspectives, providing rich insights into the phenomenon under investigation.  Qualitative approaches, primarily through intensive interviews, intentionally employ smaller sample sizes to facilitate in-depth exploration of personal viewpoints and lived experiences, which may not be achievable in larger quantitative investigations. Our sampling strategy aligns with established practices in rigorous qualitative research, where a focus on quality involves conducting numerous interviews to extract meaningful themes and draw insightful conclusions (Guest et al., 2006). The emphasis lies on achieving analytical depth rather than breadth or generalizability. We hope this clarification helps to address your concerns. Thank you again for your valuable feedback, and we remain committed to enhancing the rigor and value of our research.